# Circadian control of lung inflammation in influenza infection

Shaon Sengupta[1,2], Soon Y. Tang[2,3], Jill C. Devine[2], Seán T. Anderson [2,3], Soumyashant Nayak[2,4], Shirley L. Zhang [5], Alex Valenzuela[6], Devin G. Fisher[6], Gregory R. Grant[2,4], Carolina B. López [2,6] & Garret A. FitzGerald[2,3]

Influenza is a leading cause of respiratory mortality and morbidity. While inflammation is essential for fighting infection, a balance of anti-viral defense and host tolerance is necessary for recovery. Circadian rhythms have been shown to modulate inflammation. However, the importance of diurnal variability in the timing of influenza infection is not well understood. Here we demonstrate that endogenous rhythms affect survival in influenza infection. Circadian control of influenza infection is mediated by enhanced inflammation as proven by increased cellularity in bronchoalveolar lavage (BAL), pulmonary transcriptomic profile and histology and is not attributable to viral burden. Better survival is associated with a time dependent preponderance of NK and NKT cells and lower proportion of inflammatory monocytes in the lung. Further, using a series of genetic mouse mutants, we elucidate cellular mechanisms underlying circadian gating of influenza infection.

[1] Department of Pediatrics, University of Pennsylvania Perelman School of Medicine, Philadelphia, PA 19104, USA. [2] Institute of Translational Medicine and Therapeutics, University of Pennsylvania, Philadelphia, PA 19104, USA. [3] Systems Pharmacology, University of Pennsylvania Perelman School of Medicine, Philadelphia, PA 19104, USA. [4] Department of Genetics, University of Pennsylvania Perelman School of Medicine, Philadelphia, PA 19104, USA. [5] Department of Neuroscience, University of Pennsylvania Perelman School of Medicine, Philadelphia, PA 19104, USA. [6] University of Pennsylvania Veterinary School, Philadelphia, PA 19104, USA. Correspondence and requests for materials should be addressed to S.S. (email: SenguptaS@email.chop.edu)

Circadian rhythms constitute an innate anticipatory system with a 24-h periodicity that improves survival by helping the organism adapt to its surroundings. At the molecular level, circadian rhythms are controlled by oscillating core-clock genes, which regulate rhythmic expression of their downstream targets[1]. Many physiological processes, including immune responses[2–5], are subject to circadian regulation. Inflammation is a critical part of the immune response to influenza. While an ineffective inflammatory response impedes viral clearance, enhanced inflammation injures the host[6,7]. Due to its role in maintaining overall homeostasis, the circadian regulatory system may act to balance antiviral resistance with host tolerance, in a way that is favorable to overall survival.

Although the role of circadian regulation in systemic viral infections[8] has been described using Murid herpes virus (MHV)[9] and Vesicular Stomatis Virus (VSV)[10,11], information on respiratory viruses is limited[12]. Some in vitro work[9] and results of a cluster-randomized study testing the efficacy of influenza vaccine in older adults[13] are consistent with a role for circadian rhythms in influenza infection. However, the importance of circadian rhythms in modulating lung inflammation induced by influenza infection has yet to be systematically evaluated.

In this study, we test whether the outcome of influenza A infection (IAV), viral burden, and pulmonary inflammation were regulated by circadian rhythms. We utilize a series of genetic mouse mutants to understand the cellular mechanism underlying this regulation.

## Results

**Time of infection determines survival.** To ascertain whether the time of infection determines mortality and morbidity from influenza infection, C57BL/6J mice (male and females in approximately equal numbers) were infected intranasally (i.n.), either just before the onset of their active phase/lights off (active phase: ZT11) or just prior to the onset of their rest phase/lights-on (ZT23) with same dose of IAV (PR8 strain, 40 PFU). By convention, ZT0 refers to the time when lights turn on. The choice of these times was guided by previous reports which implicate the change from rest phase to active phase and vice versa, as the time points most likely to reflect phase reversal for the immune response to invading pathogens[14,15].

Animals were weighed and monitored daily for 2 weeks to record disease progression (Mice infected at ZT11 were always weighed and scored at ZT11, and mice infected at ZT23 were evaluated at ZT23 at serial time points following infection. This ensured that the time from infection to evaluation was identical for both groups). Mice infected at ZT11 had significantly higher mortality (71% in ZT11 vs. 15% at ZT23; $p < 0.0001$ by Mantel–Cox log-rank test) than mice infected at ZT23 (Fig. 1a). Furthermore, from day 4 post infection (p.i.) onward mice infected at ZT11 had more weight loss than mice infected at ZT23 (Fig. 1b). We also recorded clinical scores based on activity level, behavior, and respiratory distress (Supplementary Fig. 1) and found higher scores consistent with increased morbidity and mortality (Fig. 1c) in mice infected at ZT11. Furthermore, we evaluated the differences in mortality by sampling evenly around the clock (at ZT5, ZT11, ZT17, and ZT23) and found that ZT23 and ZT11 represented the two points of maximal difference in outcomes across the day (Supplementary Fig. 2); hence we used these two time points for all subsequent experiments. Since there may be differences that cannot be controlled for if the two groups are always infected at different times of day, we also used light-controlled circadian boxes to maintain mice in reverse light–dark (LD) cycles (such that at the same conventional clock time, the mice are in reverse phases. Using this model, ZT11 and ZT23

mice were infected simultaneously). The mortality and morbidity results were confirmed with mice on these reverse light–dark cycles (Supplementary Fig. 3).

These data are consistent with the hypothesis that the susceptibility to influenza-induced mortality and morbidity depends on the time of day at infection.

**The temporal gating of IAV-induced disease is circadian.** To show that the diurnal differences in mortality and morbidity were due to endogenous circadian regulation as opposed to cycling light exposures, we repeated the above experiment in mice housed in constant darkness. Here, the time that would correspond to ZT23 (or ZT11) while on a 12 h-LD cycle is referred to as CT23 and CT11. The mice were infected at either CT23 or CT11 with the same dose of IAV as above. We found that mice infected at CT11 had significantly higher mortality than the mice infected at CT23 (Fig. 1d), consistent with our previous conclusion. However, the difference between survival in ZT23 vs ZT11 (85% in ZT23 vs. 29% at ZT11) was higher than the difference between mice infected at CT23 vs. CT11 (45% in CT23 vs. 15% in CT11). We attribute this difference to the absence of light in our constant darkness experiment. To further confirm our findings, we used a genetic model of clock disruption.

**Temporal gating of IAV infection is lost in $Bmal1^{-/-}$ mice.** To confirm that the time of day difference in mortality and morbidity from influenza infection was secondary to circadian rhythms, we genetically disrupted the molecular clock by deleting $Bmal1$ and infected these mice with IAV. $Bmal1$ is the only core clock gene whose deletion is sufficient to cause arrhythmicity of locomotor activity (under constant darkness)—the hallmark of circadian disruption[16]. However, to avoid confounding by its noncircadian roles during development, we used an *ER-cre* to delete $Bmal1$ in postnatal life (6–8 weeks)[17]. Both cre[+] and cre[−] littermates were treated with tamoxifen, and infected with the same dose of IAV in constant darkness at either CT23 or CT11.

While the time of day difference in outcomes was maintained in the cre[−] littermate controls (Fig. 1e; S2; survival of 58% in CT23 vs. 22% in CT11; $p = 0.015$, by Mantel–Cox log-rank test), its magnitude was dampened compared with the WT animals in Fig. 1a, which may reflect the cumulative effect of tamoxifen[18] and the stress of frequent handling[19] for tamoxifen administration. As expected, the time of day difference was abolished among the $Bmal1^{-/-}$ animals (survival of 16% in CT23 and 25% in CT11; Fig. 1e, f). Furthermore, the overall mortality and morbidity were similar to the CT11 group of the cre[−] littermates, which has higher mortality than the CT23 (comparable with the ZT23 in WT, Fig. 1a) group (Fig. 1e, f). Taken together, this confirms that the molecular clock results in a time of day difference or circadian gating of the outcome of IAV infection and abrogation of the clock results in worse outcomes, irrespective of the time of day at infection.

**Time of infection affects viral clearance not replication.** To test if the difference in the outcomes were driven by a varying rate of viral replication, we measured viral titers in the lungs at serial time points post infection—6 h, 12 h, days 1, 2, 4, 6, 8, and 10 p.i., keeping the time from infection to tissue harvest equal for both the groups (Fig. 2a). At the earliest time point, 6 h p.i., hardly any virus was recoverable from the lungs. By 12 h, virus was detected in the lung, but titers were still comparable between the two groups at early time points—12 h, day 1 and day 2 p.i. Viral replication is known to peak around days 2–4[20], and even at those time points no difference was noticed between the two groups. By day 8 p.i., more mice infected at ZT23 had cleared the virus than

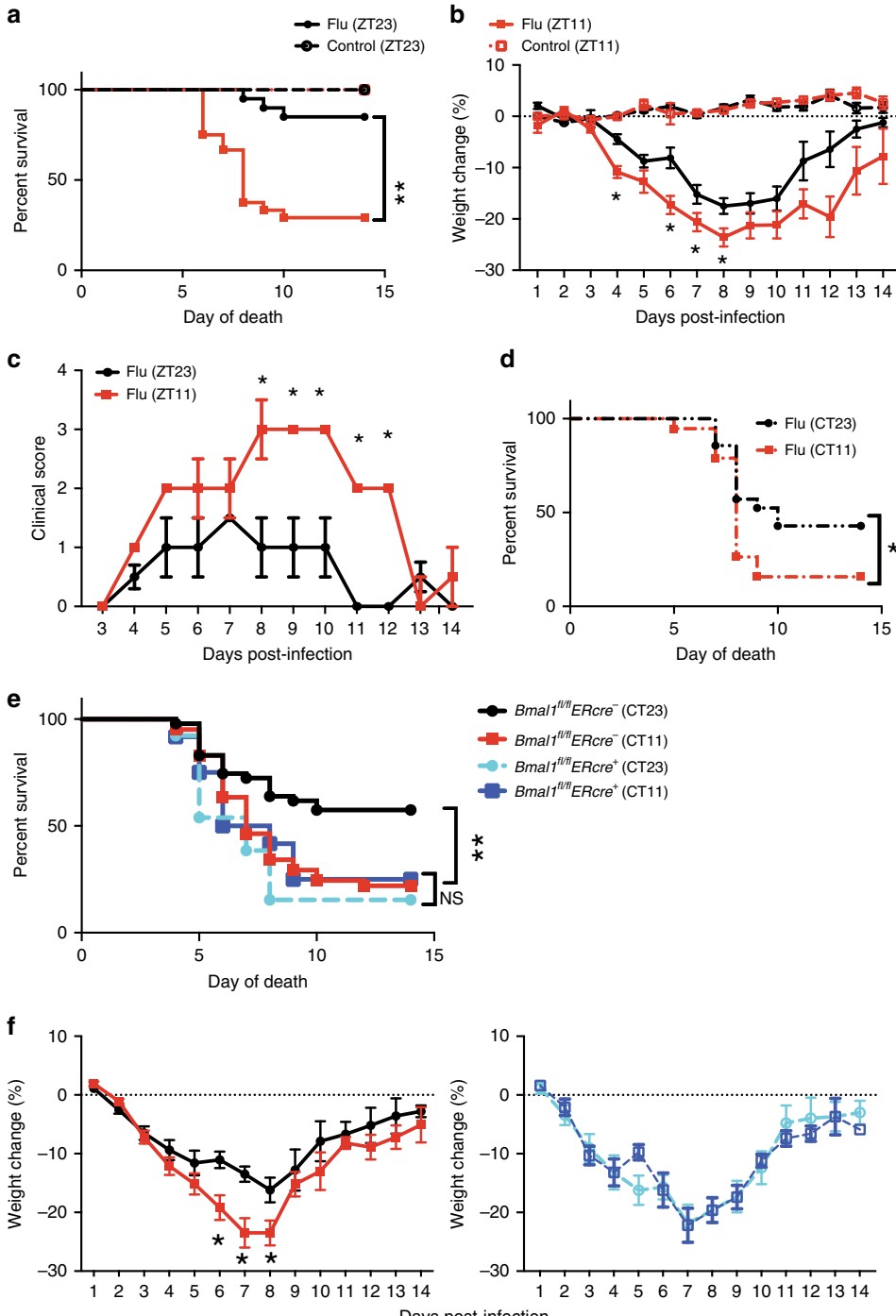

**Fig. 1** Time of infection affects survival in influenza A virus (IAV) disease. Experimental design: two groups of mice were maintained in 12 h light: dark cycles. Mice were infected with 40 PFU of IAV (PR8; H1N1) intransally (i.n.) at either the start of the light cycle (ZT23; ZT0 being the time at which light go on in a 12 h LD cycle) or at the start of the dark cycle (ZT11). Mice infected at ZT11 were always weighed and scored at ZT11 at serial time points following infection and likewise for ZT23 group. This ensured that the time from infection to evaluation was identical for both groups. **a** Survival curves are a composite of three independent experiments [total $n = 8$/control group; $n = 20$–24/IAV group, log-rank (Mantel–Cox) test, $p < 0.0001$]. **b** Disease progression is expressed as the percent of weight change following IAV infection. **c** Disease progression was also measured as clinical scores. The data represented as median ± SEM (total $n = 17$ per group; Student's $t$ test, *$p < 0.05$, **$p < 0.01$, ***$p < 0.001$). **d** Two groups of mice were maintained in constant darkness for 72 h, and were infected i.n. with 40 PFU of IAV (PR8) either at the times corresponding to start of the light cycle (CT23) or the start of dark cycle (CT11). Survival curves composite of two independent experiments [total $n = 8$–12 per group, log-rank (Mantel–Cox) test, *$p < 0.05$]. **e**, **f** Experimental design: $Bmal1^{fl/fl}$ $ERcre^+$ mice and their $cre^-$ littermates were treated with tamoxifen at 6–8 weeks of age, and acclimatized to reverse cycles of 12 h LD for 2 weeks. Thereafter, they were maintained in constant darkness for 2–4 days prior to administering IAV (PR8) at CT23 and CT11. **e** Survival (**f**) weight change trajectory [$n = 12$–13 in cre$^+$ group and $n = 41$, 47 in cre$^-$ group from three independent experiments]. Compiled data are expressed as mean ± SEM in panel **b** and **f**. Source data are provided as a Source Data file

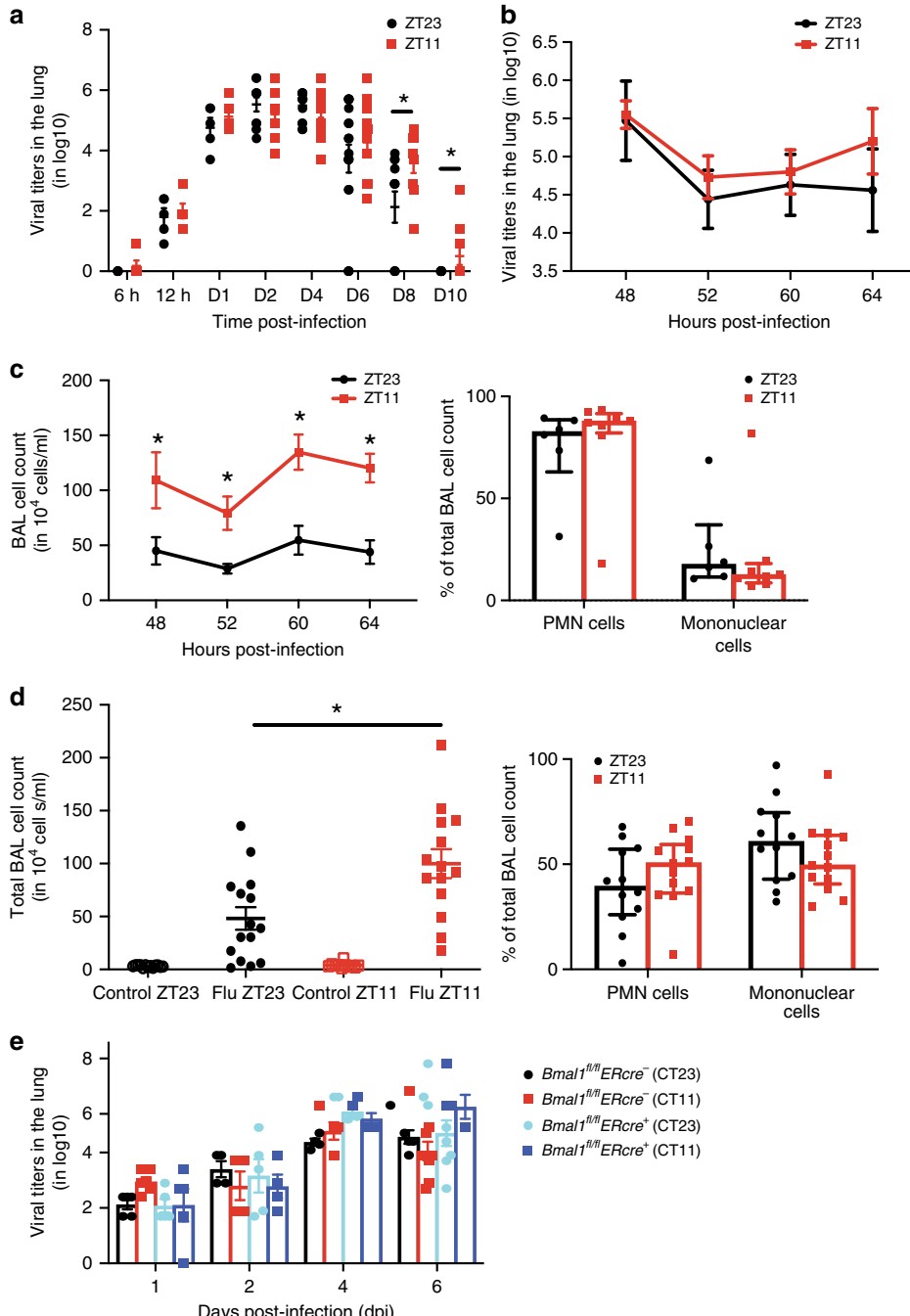

**Fig. 2** The time of infection affect late-viral clearance, not early replication. Experimental design: after infecting mice at ZT23 or ZT11, viral titers were determined in the lungs harvested at serial time points, post infection. The time from infection to tissue collection was the identical for both groups. **a** Combined data for viral titers from 6 h to 10 day post infection ($n = 5$–12 mice per group, student's $t$ test; *$p < 0.05$, ZT23 vs. ZT11; The data were pooled across 3–4 independent experiments). **b** Viral titers from ZT23 and ZT11 groups were determined at 48, 52, 60, and 64 h. **c** Bronchoalveolar lavage (BAL) was also collected at the same time points as in panel **b**, quantified and the differential was determined by staining cytospin preparations. Data were compiled from 4 independent experiments for both panels **b** and **c**, including one from reverse LD cycles (total $n = 6$–8 per time point, two-way ANOVA; *$p < 0.05$ for time of infection, NS for time of dissection). **d** Right panel: the total BAL cell count on day 6 p.i. from mice who received either IAV or PBS at ZT23 or ZT11. The data compiled from three independent experiments (total $n = 11$–13 per time point, one-way ANOVA; *$p < 0.05$, ZT23 vs. ZT11). Left panel: differential of the BAL cells from both IAV-infected groups. **e** Viral titers from *Bmal1*^fl/fl^*ERcre*+ mice and their *cre*− littermates (treated as in Fig. 1e), and samples harvested on day 1, 2, 4, and 6 dpi. (total $n = 4$–9 per time point, two-way ANOVA; $p < 0.05$ for days after infection and N.S. time of infection i.e., CT11 vs. CT23. The data were pooled across four independent experiments). The data expressed as mean ± SEM. Source data are provided as a Source Data file

those infected at ZT11. Therefore, it is unlikely that the differences in mortality and weight trajectories can be attributed to viral replication, because clearance follows rather than precedes the mortality and morbidity observed. Thus, despite inciting higher inflammation in the ZT11 group, viral clearance is delayed. Further, since several previous studies have reported higher morbidity and mortality in females[21,22], we also stratified the experiment by gender, but observed no difference in viral kinetics (Supplementary Fig. 4). We repeated this experiment with mice with $Bmal1^{fl/fl}$ $Er$cre$^+$ and their cre$^-$ littermates and found similar results, with no difference in the viral titers by genotype (Fig. 2e). This proves that the circadian control of the outcomes from IAV are not mediated by direct effects of viral replication or antiviral responses.

**Circadian gating is associated with lung inflammation**. Since there were no differences in the pulmonary viral loads between the groups infected at ZT11 and ZT23, we compared the extent of inflammation in these two groups. We collected BAL at 4–8 h intervals across 16 h from day 2 to day 3 p.i., coinciding with the peak of viral replication. While the viral titers were very similar between the two groups (Fig. 2b), the total BAL cell counts were higher in the ZT11 group across all time points, proving that infection at ZT11 promoted more inflammation, independent of the rate of viral replication or viral burden, and that this was not a function of the time of tissue collection (Fig. 2b). In fact, mice infected at ZT11 had higher total BAL cell counts even on day 6 (Fig. 2d); however, based on BAL staining and microscopy, we could not detect differences in the cell differential (Fig. 2c, d). The BAL cell count did not appear to oscillate across time (Supplementary Fig. 5), although circadian oscillations in CD45$^+$ cells have been reported in dissociated whole-lung preparations[23].

Mice infected at ZT11 had more lung injury on both days 2 and 6 p.i. (Fig. 3a, b) based on higher peri-bronchial inflammation, peri-vascular inflammation, inflammatory alveolar exudates, and epithelial necrosis. Similar, but less severe effects, were also seen with a sub-lethal dose of X31 strain of influenza virus, suggesting that these responses reflect the circadian control of influenza infection overall (Supplementary Fig. 6) rather than a strain-specific effect. Interestingly, when we challenged mice with i.n. polyinosinic:polycytidylic acid (Poly I:C), a TLR3 ligand, at either ZT23 or ZT11, the results were reversed (Supplementary Fig. 7), with mice infected at ZT23 exhibiting worse lung injury and higher total BAL cell counts. This finding suggests that the difference in outcomes in mice injected with IAV at different time points is not likely due to pathways downstream of TLR3. Finally, we also found that the time of day difference in the severity of lung histology is abolished in $Bmal1^{fl/fl}$ $Er$cre$^+$ infected with IAV (day 6 post infection), but maintained in their $Bmal1^{fl/fl}$ $Er$cre$^-$ littermates (Fig. 3c). Considered together, our results are consistent with the hypothesis that endogenous circadian rhythms determine mortality and pathology in IAV infection by modulating the inflammatory response, rather than through an impact on viral load in the lung.

**The distinct transcriptomic signature in circadian gating of IAV**. To determine whether the circadian control of IAV resulted in disparate transcriptional landscape, we undertook transcriptional analyses of whole lungs on day 6 ($n = 3$–4/group: samples were collected 6 days after infection from animals infected with either PBS or IAV at ZT11 or ZT23; total of four groups). We chose this time point because the weight loss trajectories had clearly diverged by then. However, we acknowledge that causal networks are likely to precede rather than follow weight loss, thus the differentially expressed genes and pathways discovered in our

analyses would confirm that our observed phenotype has a transcriptomic correlate rather than signal the initiating events.

Of a total 29,114 genes, 4667 had a ≥2-fold difference between the two IAV-infected groups. Although our experimental design was not set up to detect circadian rhythms in gene expression, 184 genes had a ≥ 2-fold difference from the control groups at ZT23 and ZT11 (Fig. 4a). Mice infected at ZT11 had a distinct transcriptomic profile compared with all other groups (Fig. 4b, c). As expected, innate immune pathways involving cell adhesion and diapedesis are among the most enriched and are consistent with innate immunity-mediated circadian gating of the flu infection (Fig. 4d; Supplementary Fig. 8). Further pathways enriched in the transcriptomic analyses, included various aspects of innate immunity and the "Role of hypercytokinemia/ hyperchemokinemia in pathogenesis of Influenza". We assayed the BAL from mice on day 6 p.i. and found that one cytokine IL10, known to exert anti-inflammatory action in influenza infection[24], was higher in the ZT23 group than in ZT11 animals (Fig. 3d). Surprisingly, no other differences were found either on day 6 in BAL or in the lung homogenate on days 1–6 (Fig. 3d; Supplementary Fig. 9). We speculate that the other pro-inflammatory cytokines may peak later, following the changes in transcriptome on day 6. Overall our results including the transcriptomic analyses and the literature[25,26] are consistent with the observation that a state of hyper-inflammation is induced by infection at ZT11, to which the host eventually succumbs.

**NKT, NK, and Ly6C$^{hi}$ cells: role in the temporal gating**. To identify which innate immune cells mediate the circadian difference in the response to influenza, we determined the proportion of CD45$^+$ populations in the lungs of mice infected at either ZT11 or ZT23 on days, 1, 2, 4, and 6. First, similar to our results from BAL analyses, we found that mice infected at ZT11 had a higher total CD45$^+$ cell count, which was 1.3 to 1.5 times of the count in ZT23 group, in the early phase of infection (days 1–4) (Fig. 5a). There were no significant differences, in the numbers or percentage of macrophages, neutrophils, or CD11b$^+$ DCs in the lungs, between the two groups (Fig. 5b). The absolute numbers and proportion of inflammatory monocytes (CD45$^+$SiglecF$^-$Ly6G$^-$CD11b$^+$Ly6c$^{hi}$) were higher in the ZT11 p.i. group, suggesting a state of heightened inflammation (Fig. 5c). The percentage of CD45$^+$ NK1.1$^+$LysG$^-$ cells was higher in the ZT23 group than the ZT11 on days 1–4 p.i., however, the absolute numbers were not significantly different between the two groups both in the naive as well as the flu-infected lung (Fig. 5d; Supplementary Fig. 11a). Both NK cells (CD45$^+$LysG$^-$Nk1.1$^+$CD3$^-$) and NKT cells (CD45$^+$LysG$^-$Nk1.1$^+$CD3$^+$) were higher in the ZT23 group (Supplementary Fig. S11b). Finally, using $Bmal1^{fl/fl}$ $Er$cre$^+$ and $cre^-$ littermates infected with IAV (days 1 and 2 post infection), we found that while the percentage of NK1.1 + cells were higher in cre$^-$ littermates infected at CT23 (than CT11), there were no differences noted in the $Bmal1^{fl/fl}$ $Er$cre$^+$ animals (Fig. 5e). Some recent reports have linked circadian regulation of the adaptive immune system with disease pathology[27] and lymphocyte trafficking[28], while others have not[29]. In our model, by day 8 or 10 p.i., there were no significant time-dependent differences in the total CD8 + cells or activated CD8 + cells in either the lung or mediastinal LNs (Supplementary Fig. 12).

Thus, based on these findings, we propose that individually a preponderance of NK1.1$^+$ cells may protect the host from influenza-related inflammation in the ZT23 group, while inflammatory monocytes may predispose to the enhanced inflammation in the ZT11 group. The second possibility is that they do not act alone, but rather an altered balance between NK1.1$^+$ cells and Ly6C$^{hi}$ inflammatory monocytes in the ZT23

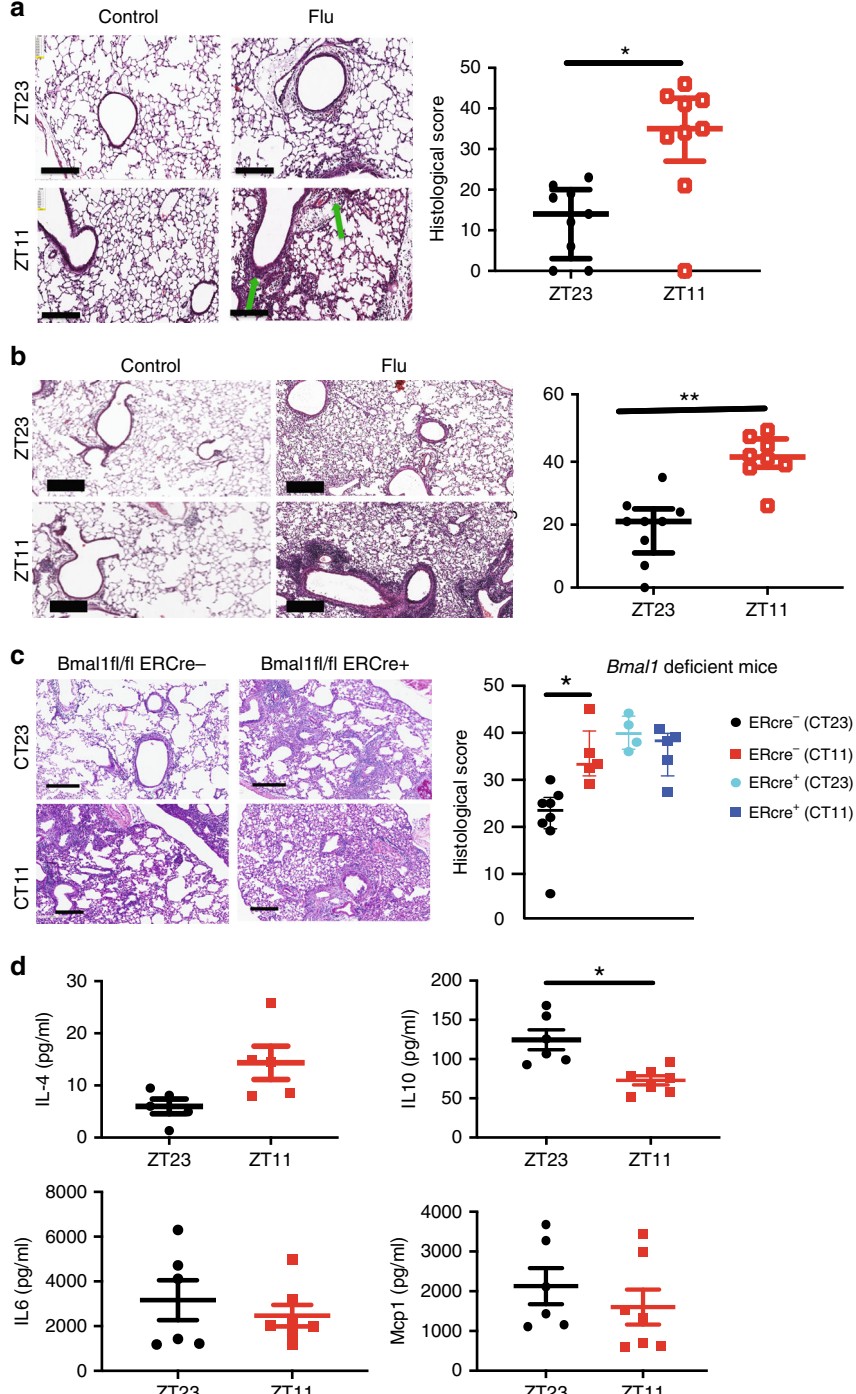

**Fig. 3** Temporal gating of IAV is associated with lung inflammation. **a** Top panel: representative micrographs of H&E stained lung sections 2 days after sham (PBS) or IAV (40 PFU) treatment (photomicrograph bar = 200 μm). Lower panel: severity of lung injury quantified using an objective histopathological scoring system by a researcher blinded to study group ($n = 5$–6/group; Wilcoxon rank-sum test; *$p < 0.05$, ZT23 vs. ZT11). **b** Top panel: representative micrographs of H&E-stained lung sections 6 days after sham (PBS) or IAV (40 PFU) treatment (photomicrograph bar = 200 μm). Lower panel: severity of lung injury quantified as above ($n = 7$–9 mice/group; Wilcoxon rank-sum test; **$p < 0.01$, ZT23 vs. ZT11). **c** Top panel: representative micrographs of H&E-stained lung sections 6 days after IAV (40 PFU) treatment of *Bmal1*^fl/fl^*ERcre*^+^ mice and their *cre*^− littermates (photomicrograph bar = 200 μm). Lower panel: severity of lung injury quantified as above ($n = 4$–8 mice/group; data as median, IQR; Wilcoxon rank-sum test; **$p < 0.01$, CT23 vs. CT11 for Cre^+ versus Cre^− animals; pooled data from two independent experiments). The data expressed as median, IQR in panels **a**–**c**. **d** Cytokine levels in BAL on day 6 post infection ($n = 6$/group. Student's *t* test; *$p < 0.01$ with post hoc correction for multiple comparisons; pooled data from three experiments). Source data are provided as a Source Data file

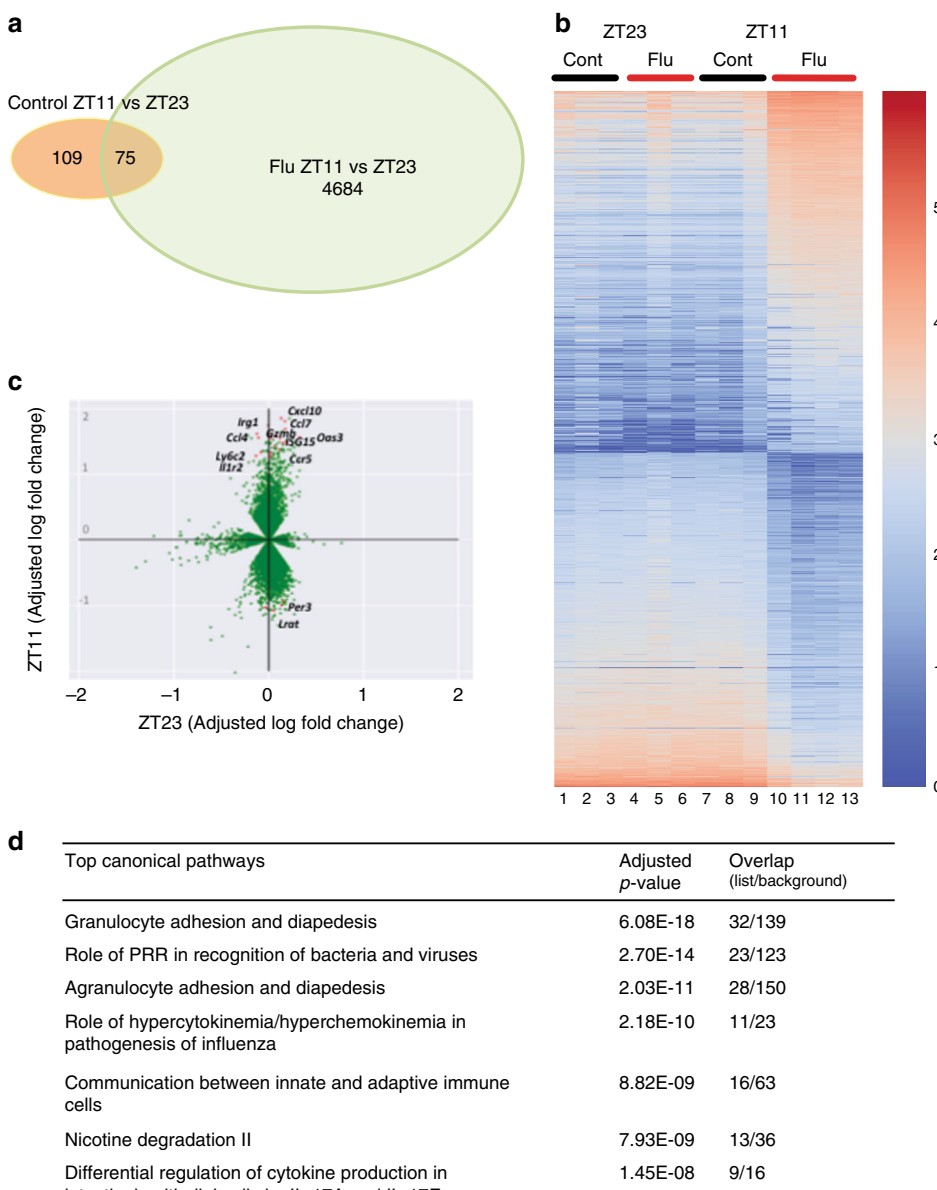

**Fig. 4** Transcriptomic analyses confirm disparate phenotype of the ZT23 and ZT11 groups. RNA samples from animals infected at ZT23 or ZT11, with either PBS or IAV, 6 days after infection were collected at ZT23 and ZT11 and used for RNA-Seq. **a** Venn diagram (sizes not to scale) depicting the number of differentially expressed genes. **b** Heatmap of the top 900 differentially expressed genes (color scheme reflects logarithmic gene expression of each group; highest in red and lowest in blue). **c** Plot of log-adjusted fold change for ZT11 and ZT23 showing directionality of the most differentially expressed genes. Flow cytometry-based enumeration of the different innate immune cell populations in dissociated lungs following IAV infection at either ZT23 or ZT11. **d** Ingenuity pathway analyses reveals the top ten (adjusted $P < 0.05$) phenotypes related to these genes. Overlap, the number of appearing genes/number of background genes. Source data are provided as a Source Data file

group may promote a pulmonary milieu, wherein inflammation is well-contained without impeding viral replication early in the course of the inflammation. To investigate their individual roles, we employed tissue-specific knock outs and depletion strategies.

**Lung and myeloid clock contributes to circadian gating.** Based on the enumeration of CD45[+] cells in the lung (Fig. 5), we investigated whether NK1.1[+] cells or inflammatory monocytes

were responsible for the circadian gating of IAV infection. There are few optimal strategies to target Nk1.1[+] cells via the cre-lox system. One model utilizes Ncr-Cre but also includes innate lymphoid cells (ILCs), which are known to be involved in Influenza pathogenesis[30]. Therefore, to determine the effect of NK1.1[+] cells, we administered an NK1.1 antibody one day prior to infection, and then infected the mice at ZT23 or ZT11 with the same dose of IAV (depletion was confirmed as in Supplementary Fig. 13). Depletion of NK1.1[+] cells, abolished the time of day

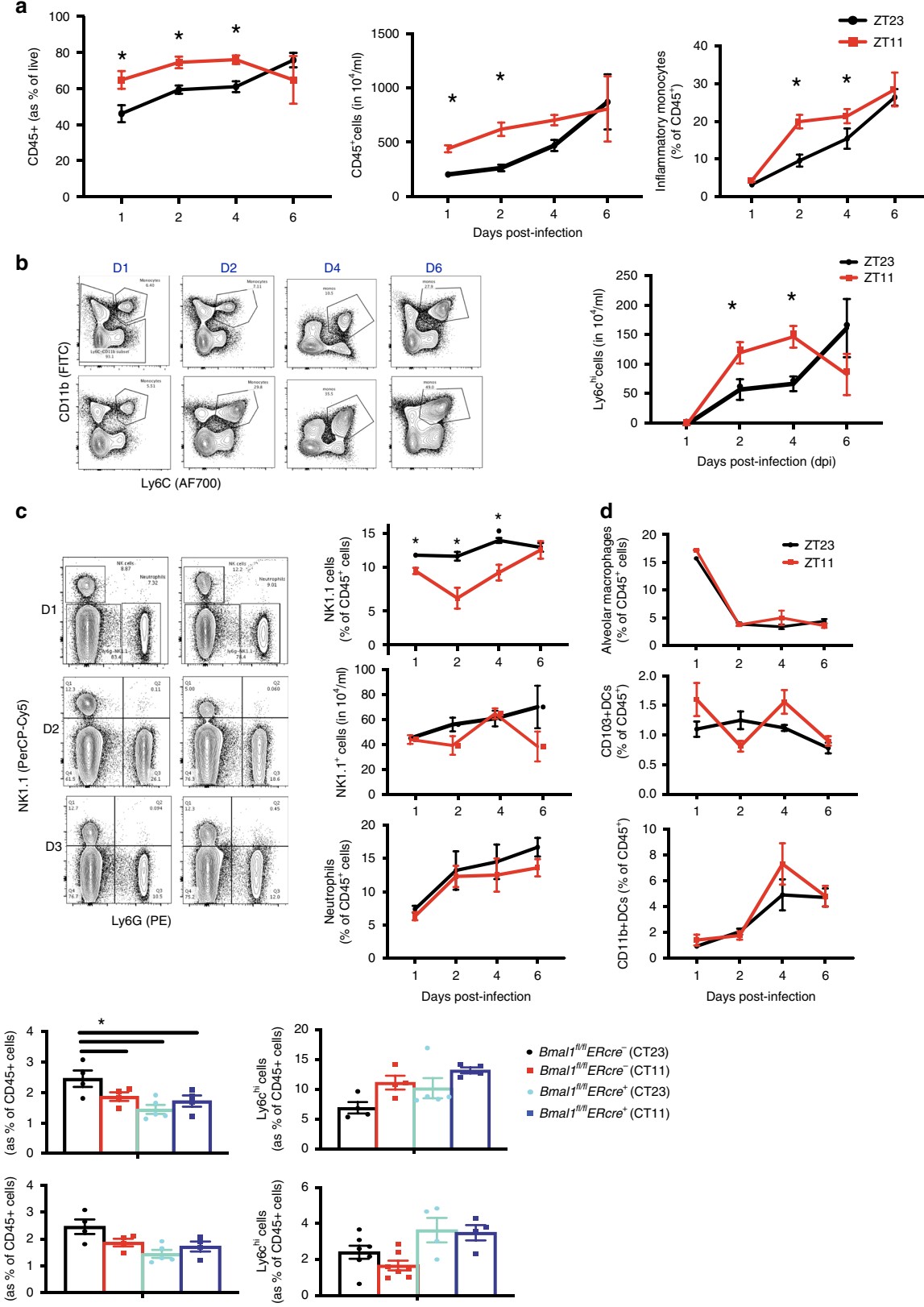

difference in outcomes and generally had outcomes comparable with the ZT11 group among WT animals that had worse outcomes (Fig. 6a, b). We used the *LysM-CreBmal1*[fl/fl] which deletes *Bmal1* in the myeloid lineage, to disrupt the clock in inflammatory monocytes. This strategy targets the entire myeloid lineage (including neutrophils and macrophages, the former being

implicated in our transcriptomic profiling), but has been employed to interrogate the effect of inflammatory monocytes. The littermate controls maintain their circadian gating (survival of 80% in CT23 vs. 25% in CT11; $p < 0.05$ by Mantel–Cox log-rank test; Fig. 6c). However, deletion of *Bmal1* in the myeloid compartment, while completely obliterating the time of day effect

**Fig. 5** Ly6C[hi] monocytes, NKT, and NK cells in temporal gating of IAV infection. The left lung lobe was digested, dissociated, and analyzed by flow cytometry. **a** CD45[+] cells (as a % of live) and CD45+ cell numbers. Two-way ANOVA, $p < 0.01$ for time of infection <0.05 for day of dissection and $p < 0.05$ for interaction. For $p < 0.05$ for time of dissection, time of infection and interaction. **b** Macrophages, two subsets of dendritic cells (CD11b + and CD103+) using gating strategies from Supplementary Fig. 10a. **c** Ly6C[hi] inflammatory monocytes with images from representative experiment. For monocytes, two-way ANOVA, $p < 0.0001$ for time of infection, $p < 0.001$ for day of dissection and N.S. for interaction. **d** Neutrophils (top panel) and absolute NK1.1[+] cells (middle panel), and % of NK1.1[+] cells (bottom panel). Two-way ANOVA. For % NK1.1[+] cells, $p < 0.05$ for time of infection, <0.05 for day of dissection, and $p < 0.05$ for interaction. For neutrophils % and NK cell numbers, no comparisons were significant. For **a–d**, representative results from one experiment are shown. Experiments were repeated with similar results three times. **e** NK cells and Ly6C[hi] cells (as % of total CD45 cells) from Bmal1[fl/fl]ERcre[+] mice and their cre[−] littermates. For NK cells two-way ANOVA, $p =$ ns for time of infection, and $p < 0.05$ for day of time of dissection in Cre[−] animals, but no difference in Cre[+] groups and $p < 0.05$ for interaction. The experiment was done once. **c–e** Using gating strategies from Supplementary Fig. 10b. The data expressed as mean ± SEM. Source data are provided as a Source Data file

(54% in CT23 vs. 50% in CT11), does not result in as high mortality or morbidity as the control CT11 group, the outcomes being intermediate in severity (Fig. 6c, d; Supplementary Fig. 7). This suggests that while inflammatory monocytes do contribute toward the circadian regulation of outcomes in IAV, they cannot alone account for the severity of the phenotype in the ERcre[+] animals.

Next, to investigate whether the lung clock contributes to the circadian gating of the outcomes, we used the CCSP-cre[+] mice, which lack Bmal1 in its club cells[31]. This epithelial clock affects outcomes in LPS-mediated lung injury via CXCL5[15]. Bmal1 deficiency in these epithelial cells removed the time of day difference in outcomes (survival of 37.5% in CT11 and 25% in CT23 among cre[+] animals; Fig. 6e; Supplementary Fig. 8). Both groups of cre[+] mice revealed enhanced mortality and morbidity which was comparable with the cre[−] littermates infected at CT11 (or WT mice infected at ZT11; Fig. 6e, f), while the cre[−] animals infected at CT23 had lower mortality (survival of 76.2% in CT23 vs. 25% in the CT11 group among cre[−] animals; $p < 0.05$ by Mantel–Cox Log-rank test).

From the above experiments, we conclude that, while NK1.1[+] cells, monocytes, and lung epithelial (club) cells all contribute toward the circadian gating of IAV infection, NK1.1[+] cell and club cells emerge as the more relevant cell types through which the molecular clock controls the severity of the host response to IAV.

## Discussion

We demonstrate here that the time of day of infection consistently determines the outcome of influenza infection. This effect is seen under 12 h LD conditions, in constant darkness, as well as in genetic models of clock disruption (ERcre[+/+]Bmal1[fl/fl]). Taken together, our results establish the role of the molecular clock in determining host response to influenza infection. Interestingly, we show that the clock does so, not by directly affecting the pathogen burden, but by altering the inflammation generated as the host fights the infection. In our model, animals had worse outcomes when infected just prior to their active phase or ZT11 (vs. just before the onset of their rest phase or ZT23) and these animals had higher BAL cell counts, more severe lung injury (on histology) and a distinct transcriptomic signature consistent with enhanced inflammation. Thus, the role of the circadian clock, in balancing antiviral clearance/resistance and host tolerance, underscores the homeostatic influence of the clock in health and disease. From an evolutionary standpoint, the heightened immune response was probably designed as an anticipatory protection of the organism from environmental threats that are likely to be encountered during the active period.

Several previous studies have found that bacterial and viral pathogens are more abundant when the host is infected at the time of day that is associated with worse outcomes, hence attributing the cause of circadian variability in response directly to the

burden of the pathogen[9,12]. Gibbs et al.[15], reported a difference in the bacterial load within 24–48 h of infection with *Streptococcus pneumonia*. More relevant to viral replication in vivo, Ehlers et al.[12] reported that antiviral activity tracked with viral nucleic acid abundance in Sendai virus infection in Bmal1[−/−] mice where the clock is disrupted. Similarly, viral replication in the host was responsible for the circadian control of infection by a luciferase-expressing Murid Herpes strain[9] in mice and for influenza in Bmal1[−/−] fibroblasts[32]. The difference between our results and those of Edgar et al.[9] and Ehlers et al.[12] are likely attributable to the model of infection used and to the method of determining viral load, respectively. While there are other methods of determining viral load, viral titers are the gold standard[33]. Viral loads measured as viral RNA expression, for example, may be confounded by other factors, such as the presence of defective viral genomes (DVG). Edgar et al.[9] did not report an in vivo model of IAV infection; they infected mouse embryonic fibroblasts with a luminescence-tagged virus, where the contribution of different lung cells and CD45[+] cells to viral replication were not assessed. In vivo, IAV titers peak around day 2–4 in mice. We found no difference between the groups from 6 h to day 6 post infection, either in WT animals or in mice where the clock has been genetically disrupted by Bmal1 deletion. Furthermore, our results are consistent with existing reports that mortality in influenza infection may be caused by extensive activation of immune pathways[34], rather than viremia or extrapulmonary dissemination[34–36].

In elucidating the cellular mechanism underlying the circadian regulation of influenza-induced lung inflammation, we systematically determined the various cell populations in the lung across serial time points following infection. Bmal1[−/−] mice have been used extensively in the literature to interrogate clock-dependent mechanisms given the nonredundant role of this transcription factor in circadian regulation[5,9,12,14,15,37,38]. However, phenotypes consequent to its embryonic deletion reflect both circadian and noncircadian roles[17]. Specifically, clock does not become available to dimerize with Bmal1 and allow circadian function until late in pregnancy[39]. We have used the ER-cre[+] Bmal1[fl/fl] mice to bypass phenotypic consequences of off-target effects of Bmal1 during pregnancy. We suspect that some of these off-target effects may have resulted in the differences between our conclusions and those of other reports based on conventional knock outs. The two groups of WT mice infected at ZT11 and ZT23 were differentiated by an abundance of NK1.1[+] cells and reduced inflammatory monocytes in the ZT23 group. Therefore, we investigated the roles of these cells together with that of lung epithelial cells in mediating the circadian-regulatory control of influenza virus-induced lung inflammation. While circadian rhythms in all three cell types, contribute to the time of day effects, the severity of infection was mostly regulated by clocks in lung epithelium and the NK1.1[+] cells. However, our transcriptomic profiling was heavily biased toward innate immune pathways

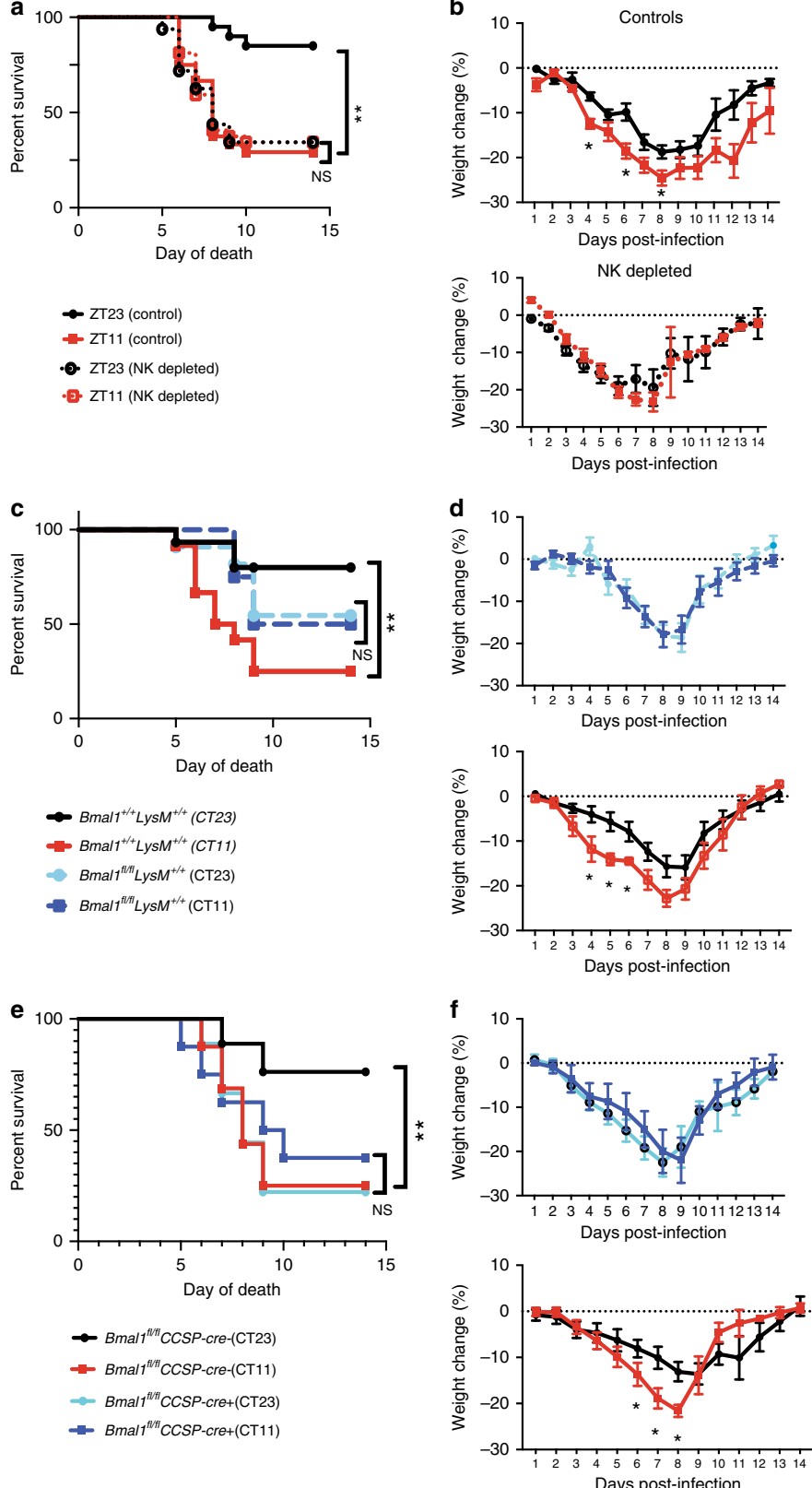

and hyper-inflammation, and did not reflect the pulmonary epithelial response to inflammation directly. We believe, this was secondary to the fact that on day 6 post infection, the whole-lung transcriptome is dominated by an overabundance of immune cells, which potentially dwarfs contributions from other cellular players.

Additional targets are also of likely relevance to the circadian regulation of inflammation. For example, the importance of neutrophils and Cxcl5 in mediating the response to LPS in the lung was revealed in mice, where *Bmal1* was deleted specifically in the ciliated club cells of the lung (*CCSP-Bmal1*). In response to Listeria, *Nyugen* et al.[3] noted a preponderance of inflammatory

**Fig. 6** Immune and lung clocks contribute to circadian gating of IAV infection. Experimental design: C57bl6 mice were maintained in reverse cycles of 12 h LD for 2 weeks. Thereafter, Nk1.1 antibody administered to deplete Nk1.1+ cells, and one day later the animals were infected with IAV (PR8) at ZT23 and ZT11. **a** Survival. **b** Weight change trajectory (N = 20–32/group form three independent experiments). Experimental design: Bmal1fl/flLysMcre+ mice (mice lacking Bmal1 in the myeloid cells) and their Bmal1+/+cre+/+ littermates were acclimatized to reverse cycles of 12 h LD for 2 weeks. Thereafter, they were maintained in constant darkness for 1 week prior to administering IAV (PR8) at CT23 and CT11. **c** Survival. **d** Weight change trajectory (n = 11–15/group from three independent experiments). Experimental design: Bmal1fl/flCCSPcre+ mice (mice lacking Bmal1 in club cells of the lung epithelium) and their cre− littermates were acclimatized to reverse cycles of 12 h LD for 2 weeks. Thereafter, they were maintained in constant darkness for 1 week prior to administering IAV (PR8) at CT23 and CT11. **e** Survival (**f**) weight change trajectory (n = 9 in cre+ groups and n = 9–16 in the cre− group in two independent experiments). Survival curves composite of 2–4 independent experiments (log-rank (Mantel–Cox) test, *p < 0.05). The data expressed as mean ± SEM in panels **b**, **d**, **f**. Source data are provided as a Source Data file

Ly6c^hi monocytes in the spleen of $Bmal1l^{fl/fl}Lyz2^{Cre}$ mice. More recently, Poullaird et al. noted increased neutrophils in response to LPS in $Rev-erbα^{−/−}$ mice at ZT0, but not at ZT4[40]. While it is probable that neutrophils may serve as the effector cells for the circadian regulation of TLR4 signaling pathways[15,40,41], this mechanism does not underlie the response to viruses. In conventional global $Bmal1^{−/−}$ mice infected with Sendai virus, there were no significant differences in neutrophils, alveolar macrophages, or dendritic cells in the lung on days 1 and 5 following infection[12]. In some studies, the effect of Bmal1 on viral nucleic acid expression[9] or mortality[37] was described, but specific cellular mediators in the host's immune repertoire were not examined. In mechanisms specific to circadian regulation of viral infections, oscillation of the pattern recognition receptor, TLR9 has been reported[42]. These differences likely reflect a pervasive impact of the circadian network on the diverse elements of the innate immune system.

NK cells have direct cytolytic activity toward virally infected cells and accumulate at the site of infection, typically peaking around days 4–5 p.i[43,44] which is consistent with our results. Mice lacking NCR1, the predominant activating NK cell receptor succumbed to IAV early on, supporting the protective role of activating NK receptors in IAV[45]. Furthermore, NK cells and cytolytic function have been shown to be under circadian control in both humans[46] and animal models[47–49]. In splenocytes obtained from $Per1^{−/−}$ mutant mice, the rhythms of cytolytic activity, cytokines and cytolytic factors (such as granzyme B and perforin), and gene expression were significantly altered[48,49]. In the lungs of rats exposed to chronic jet lag, NK cytolytic activity was suppressed and tumor growth was stimulated[47]. Thus, the literature is consistent with a role for NK cells in the circadian regulation of the immune system, although not much is known about NKT cells in this context. NKT cells may influence outcome of IAV infection by limiting the migration of inflammatory monocytes[50].

In conclusion, our work demonstrates that time-dependent regulation of influenza infection and its consequences are mediated by circadian regulation of host tolerance pathways and not directly through viral replication. This temporal difference in outcomes based on time of inoculation is consistent with recent trials of vaccination that demonstrate that time of day affects antibody responses[13,51,52]. However, the findings should have broader relevance for other respiratory pathogens and circadian regulation of host–pathogen interaction. A body of work, both mechanistic[46] and epidemiological have shown that shift workers, who experience circadian disruption are at increased risk for health issues[53], including metabolic syndrome[54], cardiac diseases[55], and cancer[56,57]. Based on our results here, we would extend these possibilities to more acute conditions such as respiratory infections. We also suspect that the role of understanding and harnessing circadian regulation in disease states is further underscored by deliberate changes in our lifestyle wherein social jet lag is the normative, rather than the deviant. Finally, we speculate that perturbation of circadian rhythms in intensive care units (whether through lighting, noise, disruptive timing of food, clinical assessments, or medications) may all potentially worsen the inflammation in patients afflicted with respiratory pathogens.

## Methods

**Mice, virus, and infection.** Specific pathogen-free 8–12-week-old C57bl/6J mice were purchased from Jackson Labs. For influenza infections, mice were lightly anesthetized with isoflurane and infected intranasally (i.n.) with 40 PFU of PR8 strain or $10^4$ PFU of X31 strain of influenza virus, respectively, in a volume of 40 μl. For serial evaluations (for weights, scores, titers, and cell counts), animals were assessed at the same interval from the time of infection by researchers blinded to the study group that these animals were allocated to. Once animals had lost >20% of their body weight, the scoring and weights were increased to twice daily in most cases. All animal studies were approved by the University of Pennsylvania Institutional Animal Care and Use Committee and met the stipulations of the Guide for the care and Use of Laboratory animals.

**Genetic mouse mutants.** To generate inducible Bmal1 knockout (iKO) mice, 2-month-old (unless specified) $Bmal1^{fl/fl}$-ERcre+ mice were treated with 5 mg (in 50 μl) of tamoxifen via oral gavage, each day for 5 consecutive days. Tamoxifen was reconstituted to 100 mg/ml solution with ethanol and corn oil, and thawed at 55 °C prior to administration. Cre− $Bmal1^{fl/fl}$ littermates treated with tamoxifen served as controls. They were acclimatized to reverse cycles and then exposed to constant darkness for 3–4 days before being infected with PR8 IAV at CT23 and CT11. $Bmal1^{fl/fl}$ were crossed with LysM-Cre+/+ knock-in mouse line, which express Cre recombinase under the control of the LysozymeM promoter to produce progeny that have Bmal1 excised in the myeloid lineage (monocytes, macrophages, and granulocytes). $Bmal1^{fl/fl}$ LysM-Cre+/+ were compared with littermate controls, $Bmal1^{+/+}LysM-Cre^{+/+}$ as described previously[14]. CCSP-cre mice were a kind gift from E.E. Morrissey (University of Pennsylvania) and were crossed into the $Bmal1^{fl/fl}$ line and resulted in a deletion of Bmal1 in club cells in the lung[31]. Both males and females were used in all above experiments in approximately equal numbers.

**Nk1.1+ cell depletion.** For these experiments, each mouse received 200 μg Nk1.1+ antibody (BioXcell, InvivoMab, clone PK136, cat. no. BE0036) by intraperitoneal injection 24 h before the IAV infection. Control mice received PBS[58]. The efficacy of the depletion strategy 24 h following administration in both males and females was used in all above experiments in roughly equal numbers.

**Viral titration.** Lungs were harvested at different time points following infection, as indicated in the specific experiment. Lungs were extracted, homogenized in PBS–gelatin (0.1%), and frozen for preservation. The presence of influenza virus was evaluated using MCDK cells (gift from Scott Hensley's group: originally purchased from ATCC, cat no. PTA-6500) with 1:10 dilutions of the lung homogenates at 37 °C. After 1 h of infection, 175 μl of media containing 2 μg/ml trypsin was added, and the cells were further incubated for 72 h at 37 °C. A total of 50 μl of medium was then removed from the plate, and tested by hemagglutination of chicken red blood cells (RBCs) for the presence of virus particles. The hemagglutination of RBCs indicated the presence of the virus.

**Flow cytometry.** Lungs were harvested after PBS perfusion through the right ventricle. The lungs were digested using DNAse II (Roche) and Liberase (Roche) at 37 °C for 30 min. Dissociated lung tissue was passed through a 70 -μm cell strainer, followed by centrifugation and RBC lysis. Cells were washed and re-suspended in PBS with 2% FBS. (Details of the used antibodies are in Supplementary Table 1). In all, $2–3 \times 10^6$ cells were blocked with 1 μg of anti-CD16/32 antibody, and were stained with indicated antibodies on ice for 20–30 min. No fixatives were used. Flowcytometric data were acquired using FACS Canto flow cytometer and analyzed using FlowJo software (Tree Star, Inc.). All cells were pre-gated on size as singlet live cells. All subsequent gating was on CD45+ in the lung only. Neutrophils were identified as

live, CD45$^+$, Ly6G$^+$ cells. Ly6C$^{hi}$ monocytes were identified as live, CD45Ly6G$^-$ Ly6C$^{hi}$CD11b$^+$ cells. NK cells were identified as CD45$^+$Ly6G$^-$LysC$^-$NK1.1$^+$ cells. In some experiments, indicated in the figure legend, an exclusion gate for neutrophils and T cells (Ly6G, CD4 and CD8) was applied. Alveolar macrophages were identified as CD45$^+$ Ly6G$^-$ Siglec F$^+$; DCs were identified as live, CD45$^+$Ly6G$^-$SiglecF$^-$ CD11c$^+$MHCII$^+$ cells and further classified into CD103$^+$ conventional DCs or CD11b$^+$ DCs. Day 6 onwards, T cells were identified in mediastinal LNs as CD45 +, either CD4 + or CD8 + cells. Activated cells were further differentiated as CD44$^+$, CD62L$^{lo}$.

**Histology and BAL cytology**. Flu infected and control mice were euthanized by CO$_2$ asphyxiation, and their tracheas cannulated with a 20 G flexible catheter (Surflo, Terumo, Philippines). The lungs were gently lavaged with 600 µl of PBS in four passes. The supernatant from the first pass was collected and used for further analyses. The cells from all four passes were pooled and re-suspended in 1 ml of PBS and counted using a Nexcelcom cell counter. Lungs were fixed by inflation with 10% buffered formalin at 20 mm H$_2$O of pressure, paraffin embedded, and stained with H&E stain and PAS. Stained slides were digitally scanned at ×63 magnification using an Aperio CS-O slide scanner (Leica Biosystems, Chicago IL). Representative images were taken from scanned slides using Aperio ImageScope v12.2.2.5015 (Leica Biosystems, Chicago, IL). The histological and cytological scoring were performed in a blinded fashion. Numerical codes were used to identify these slides during the scoring. Once all the data were recorded, the identity was unmasked and final analyses undertaken according to the study group.

**Statistics**. All statistical analyses were performed using STATA 11.0 and Graph-Pad (Prism). Unpaired *t* test or ANOVA was used for normally distributed data, while Mann–Whitney was used for data without a normal distribution and for discrete scores (for lung histology). Bonferroni corrections were used for multiple comparisons.

**Statement on rigor and reproducibility**. All studies were done using animals from Jackson Labs and animals from in-house breeding. The background strain of each genetically modified animal has been specified, and controls were cre$^-$ littermates on that same background. Reported findings are summarized results from three to six independent experiments.

**RNAseq**. RNA was extracted from whole-lung homogenate using the RNA MiniElute kit, as per the manufacturer's protocol. QC was performed, and only samples with RINs >7 were used for sequencing. Library preparation was performed on 400 ng DNAse-treated RNA using the Illumina Truseq kit. Sequencing was done using HiSeq 2500 sequencer (Illumina) housed at BGI, CHOP to generate 2 × 100 strand-specific paired-end reads. We obtained 30–50 mi pairs of reads per sample. Samples were aligned to a mouse reference genome on an in-house resampling-based normalization and quantification pipeline[59] and compared with existing gene annotations (ENSEMBL) to identify novel loci and isoforms. Differentially expressed genes were identified using a false discovery rate-based control for multiple testing. Finally, Ingenuity and GSEA were used to assess fully effects on key pathways and mediators.

**Transcriptomic analysis**. The RNA-Seq reads were aligned to the mouse genome mm10.GRCm38.p5 using STAR version 2.5.3a (Dobin et al., 2013)[60]. Following alignment, the normalization and quantification procedures were performed with the PORT version 0.8.2a-beta pipeline (http://github.com/itmat/Normalization). Gene-level quantification was done by PORT with Ensemblv90 annotation. The goal of the transcriptomic analysis was to evaluate differential response to the IAV infection in the ZT23 group versus the ZT11 group. Since the comparison is across different time points, several genes are differentially expressed simply by virtue of the circadian rhythms or other time-dependent effects. To account for this, we normalized the differential expression in the IAV-infected group at each time point for the control group from the same time point.

A *p*-value based two-way ANOVA analysis to extract interaction effects is generally considered unreliable with just three replicates per condition. However, achieving a meaningful ranking of genes that informs a powerful pathway enrichment analysis is sufficient for our purposes. A detailed study outlining our systematic approach is described to find the optimal value of pseudo-count (≈20) for an adjusted fold-change measure to rank genes by expression values in RNA-Seq data is currently under review as a method-based paper. The top 900 genes with difference of adjusted log10 fold-change >0.67 (corresponding to about fivefold change in the adjusted fold changes) were used for pathway enrichment analysis. Pathway analyses were performed with ingenuity IPA. Overrepresentation enrichment analysis was performed on the list of genes exhibiting differential change across the two groups (from am to pm) using the functional enrichment analysis webtool WebGestalt (http://www.webgestalt.org/) in the three main gene ontology (GO) categories: Biological Process, Molecular Function, and Cellular Component. The generated bar charts of the intersection of our list of genes with the total list of genes.

**Reporting summary**. Further information on research design is available in the Nature Research Reporting Summary linked to this article.

## Data availability
Sequencing data from the experiment reported in Fig. 3 have been submitted to and are freely available at the Gene Expression Omnibus (GEO) and the accession number is GSE117029. The source data underlying all other figures are provided as a Source Data file and are available from the corresponding author on reasonable request.

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

## Acknowledgements

We are thankful to members of the FitzGerald Lab–S. Teegarden for help with animal breeding and general lab management; Yasmine Issah and Amruta Naik for general animal husbandry and lab management; G. S. Worthen, A. Sehgal, and T. Kambayashi who were part of the Sengupta's K08 committee and to T. Kambayashi for advice on NK cell depletion. This work was supported by the NHLBI-K08HL132053 (S.S.), NICHD-K12HD043245 (S.S.), a Maturational Human Biology grant from the Institute of Translational Medicine and Therapeutics, University of Pennsylvania (S.S.), NIH/NIAID R01AI134862 and R01AI137062 (C.B.L.) and NIH/NCRR RR023567 (G.A.F.). G.A.F. is the McNeil Professor of Translational Medicine and Therapeutics and a senior advisor to Calico Laboratories.

## Author contributions

S.S. and G.A.F. conceived the project; S.S., C.B.L., and G.A.F. designed the experiments; S.S., J.C.D., S.Y.T., S.T.A., S.L.Z., A.V., and D.G.F. performed the experiments and the collected data. S.N., G.R.G. and S.S. analyzed the transcriptomic data. S.S. and S.L.Z. analyzed the flow-cytometry data. S.S. wrote the original draft with the help from G.A.F. G.A.F. supervised research activities.

## Additional information

**Competing interests:** The authors declare no competing interests.

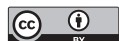

