## [Peer Review File · Nature Communications]

Reviewers' Comments:

Reviewer #1:

Remarks to the Author:

Review for NCOMMS-18-37599-T

In this report, Sengupta et. al investigated the relationship between circadian rhythms and influenza severity in mice. They found that mice infected around lights-on (ZT11) developed a more severe acute viral illness compared to mice infected proximate to lights off (ZT23). More severe infection at ZT11 was associated with: 1) delayed resolution of viremia; 2) elevated BAL leukocyte counts; 3) worsened bronchiolitis on pathologic specimens, 3) increased infiltration of inflammatory monocytes; and, 4) a depression in NK/NKT cell counts during peak viremia. Temporal variation in influenza severity appeared to be sensitive to multiple perturbations, including NK cell depletion and conditional deletion of *bmal1* in either myeloid cells or club cells. The authors concluded that circadian rhythms influence the outcome of influenza infection by regulating the inflammatory response during acute viral illness. This contrasts with a prior paper that suggested clock genes might directly regulate influenza virus replication.

Overall, this is an interesting paper on circadian biology as it relates to an important human pathogen (influenza). Viewed in the context of prior literature on this topic, the paper is novel in its focus on circadian control of lung inflammation during acute influenza infection. A technical strength of the paper is the inclusion of conditional and tamoxifen inducible *bmal1* mice.

The drawbacks (which are addressable) include: 1) some missing control data; 2) no cytokine validation that mice infected ZT11 are more "inflamed" than those infected at ZT23; and, 3) the narrative ends on an awkward note because the observations presented in the paper do not always connect. For example, NK cell number is proposed to mediate circadian effects during influenza infection (Fig. 4), but this idea is not tested in conditional *bmal1* knockout mice where time-of-day effects are abrogated (Fig. 5).

Specific Comments

Figure 1

1) Although the authors provide a reason for the selection of ZT11 and ZT23 as timepoints to infect, sampling evenly around the clock should be done to map a circadian response for at least one readout. This could be done by recording weight loss at 7-8 dpi when infections are conducted at ZT5, ZT11, ZT17, and ZT23.

2) It would be better to show the confirmation experiment using reversed light cycles as a supplemental figure rather than state data not shown (line 83).

3) Regarding panels 1E and F, mice are reported to be kept in constant darkness for 1 week prior to infection at different CTs. These are valuable experiments but keep in mind that because of free running, mice in this experiment likely experienced a 3.5 hour phase advance in the week leading up to influenza infection. Thus, the time points being compared are more akin to ZT7.5 and ZT19.5 (assuming an intrinsic period of 23.5 hours).

Figure 2

1) It would be better to show the gender segregated data for influenza infection as a supplemental figure panel rather than reporting it as data not shown (line 138).

Figure 3

1) This figure would be enhanced by some targeted cytokine measurements to validate the interpretation of gene expression data that ZT11 infections lead to lung hyperinflammation.

Figure 4

- 1) For panel 4B please provide alveolar MAC counts as a proportion of live cells. Influenza infection ought to lead to a depletion of this cell type based on prior literature, not an increase.
- 2) For panels 4D and 4E please include NKT and NK cell counts for uninfected mice at ZT11 and ZT23. Without this baseline data it is hard to interpret the difference in cell counts observed at on the first day post-infection.
- 3) Please provide control data demonstrating the extent of NK cell depletion by antibody treatment at ZT23 vs ZT11 (based on panel 4D, 1 dpi appears to be the time with the greatest differential cell count).

Figure 5

- 1) The paper would be enhanced by checking NK/NKT/inflammatory monocyte cell number in the conditional *bmal1* knockout lines and/or the *bmal1* iKO mice. This would mechanistically tie together the observations in Figure 4 and Figure 5 and make for stronger conclusions. As the paper currently stands, the idea that NK cell number mediates circadian effects during influenza is not validated and is based on a single experimental approach.

Miscellaneous

- 1) Line 152: Can you confirm reference 22 is correct for citing circadian variation in lung leukocyte trafficking?

Reviewer #2:

Remarks to the Author:

Summary

There is emerging evidence that circadian rhythms pervade many physiological responses in many tissues, including, importantly, inflammatory responses to pathogens. It is of interest to understand the immune mechanisms by which circadian rhythms affect outcomes. In a murine model of influenza A virus infection, the authors conduct a detailed investigation of the effects of timing of initial inoculum with respect to the circadian rhythm. They provide evidence that initial infection just before the start of the active period leads to higher mortality. They then dissect the underlying mechanisms to show this is not due to a difference in viral titre but an overexuberant inflammatory response, specifically including increased Ly6chi inflammatory monocytes and decreased NK1.1+ NK cells and iNKT cells. Finally, using specific deletion of clock genes in myeloid cells or pulmonary club cells they show outcomes are influenced both by circadian rhythms in myeloid cells and in epithelial cells.

This paper addresses an important, topical question and is well written, with a logical series of elegant experiments.

Major criticisms

1. The data presented appear solid, with reasonable numbers of mice and replications of experiments. However the manuscript does not mention blinding. In particular this can affect judgements about when to cull marginal animals with severe disease, and even more so clinical scoring undertaken in Fig 1C. Please state whether any blinding was used in the study (lack of blinding should not preclude publication, but should be acknowledged as a limitation). Please state whether any blinding was used for the histological scoring, as would be standard).
2. Related to this, the key foundational observation is expressed in Figure 1. Accurate weighing is essential for this readout. The mice were weighed once daily and so there may be some intrinsic diurnal variation in weight which could tend to bias risk of mortality in favour of one group more than

another (one group will be weighed at the nadir of their daily weight variation). In the ideal experiment the mice would have been weighed twice daily, in a blinded fashion. Please discuss, and/or provide evidence of the magnitude of diurnal variation in weight during illness.

3. The transcriptomic experiment is potentially interesting, but the analysis performed is rather limited leading to bland/uninformative conclusions. At least tables of differentially expressed genes should be available in online supplement. Would suggest using GSEA for signatures of specific pathways of interest, eg TLR signalling. Indeed GSEA is mentioned in the methods, but no analysis is shown.

4. Discussion: this work has significant potential clinical relevance eg for shift workers, management of flu pandemics, likely extension to other respiratory pathogens etc. Currently this is restricted to two sentences: please expand discussion of the likely implications of these findings.

Minor criticisms

1. Results: line 65 specify dose of PR8 here, as well as in methods.

2. Results: line 83 and line 138 state 'Data not shown'. Please show data in an online supplement or remove.

3. Results line 96, a change in the magnitude of an effect cannot be attributed to a sample size. Indeed might it not be more probable that this difference between models was due to an effect of light signal. Please change this explanation / if you believe this is a sample size problem, suggest perform further replicates.

4. Results paragraph beginning line 169. Please specify more about the design of this experiment: eg number of replicates (it is mentioned in methods but should also be mentioned here)

5. Results line 195: qualify the magnitude: '...at ZT11 had slightly higher total CD45+...'

6. Results line 200 and following: I wonder about correction for multiple comparisons. How many cell types were compared? Was there an a priori hypothesis? What is the risk here of Type I statistical error?

7. Discussion line 261 ff: Could you speculate on what might be the evolutionary advantage of the clock signal enhancing inflammation at certain periods.

8. Line 276: Please add the citation numbers after 'Edgar' 'Ehlers' line 284, please add relevant citation for this statement.

9. Minor typos: line 196 remove extraneous comma, 409 'were' not 'was'

10. Figure 2A: Legend: was this a single iteration of the experiment or were there more than one replicate performed?

11. Figure 2: red squares are hard to differentiate when overlapped. Suggest use diamonds instead, and make the median lines more visible (eg red group could have black summary statistics)

12. Figure 2C and 2D, 4D: bars hide data. Please show individual data points, eg as an overlay over the bars, rather than solely summary statistics.

13. Figure 3A: Specify if there was any blinding

14. Figure 4A: y axis should stop at (or a little above) 100%, not to 150%

15. Figure 4B: the gating looks messy: why are there black gates and red gates. Which was actually used for the analysis? Please show only the relevant data, or if separate gates were used for ZT23 and ZT11 groups, please explain

16. Fig 4B and 4C: axes are untidy and could be better formatted.

17. Methods: statistics section needs a statement on reproducibility (ie number of replicates of experiments)

18. References: are incomplete for refs 10,13,38,44

Overall appraisal

An interesting manuscript presenting a systematic and logical series of experiments.

Reviewer #3:

Remarks to the Author:

A manuscript by Sengupta et al described the effects of circadian rhythms on influenza virus infection. The authors found differences in influenza-induced morbidity and survival, delayed viral clearance in ZT11 mice, associated with increased lung inflammation and pathology. Differences in NK1.1 and monocyte percentages are reported.

These are interesting findings but the data need to be re-analysed before any conclusions can be made. My main concern is that some of the conclusions are based on either small differences in numbers or frequencies alone:

Fig 4A numbers: What are the actual differences in cell numbers for d1, d2 and d3 significant differences? It seems to me as perhaps 80 cells? Would these be physiologically significant differences? How many mice were analysed? How many times was the experiments repeated? Showing pulled data across experiments is needed to draw conclusions here.

Fig 4B: numbers need to be shown for inflammatory monocytes to support the conclusions. Also, statistical data should be checked as d6 does not look statistically significant.

Fig 4C: numbers should also be shown as this is what matters with respect to inflammation at the site of infection.

Fig 4D: also numbers need to be shown. NKT cells: significant differences are shown for 0.2%. Is this physiologically significant? Are the numbers different?

Other comments:

Figure 1: the authors mention that infecting mice in reverse light-dark cycles confirm results in Fig 1. These are very important controls as should be shown in Fig. 1.

We thank all the reviewers for their comments and questions. These have helped up improve the quality of the submission. Our point-by-point response is as below.

Reviewer #1 (Remarks to the Author):

Review for NCOMMS-18-37599-T

In this report, Sengupta et. al investigated the relationship between circadian rhythms and influenza severity in mice. They found that mice infected around lights-on (ZT11) developed a more severe acute viral illness compared to mice infected proximate to lights off (ZT23). More severe infection at ZT11 was associated with: 1) delayed resolution of viremia; 2) elevated BAL leukocyte counts; 3) worsened bronchiolitis on pathologic specimens, 3) increased infiltration of inflammatory monocytes; and, 4) a depression in NK/NKT cell counts during peak viremia. Temporal variation in influenza severity appeared to be sensitive to multiple perturbations, including NK cell depletion and conditional deletion of *bmal1* in either myeloid cells or club cells. The authors concluded that circadian rhythms influence the outcome of influenza infection by regulating the inflammatory response during acute viral illness. This contrasts with a prior paper that suggested clock genes might directly regulate influenza virus replication.

Overall, this is an interesting paper on circadian biology as it relates to an important human pathogen (influenza). Viewed in the context of prior literature on this topic, the paper is novel in its focus on circadian control of lung inflammation during acute influenza infection. A technical strength of the paper is the inclusion of conditional and tamoxifen inducible *bmal1* mice.

The drawbacks (which are addressable) include: 1) some missing control data; 2) no cytokine validation that mice infected ZT11 are more "inflamed" than those infected at ZT23; and, 3) the narrative ends on an awkward note because the observations presented in the paper do not always connect. For example, NK cell number is proposed to mediate circadian effects during influenza infection (Fig. 4), but this idea is not tested in conditional *bmal1* knockout mice where time-of-day effects are abrogated (Fig. 5).

Specific Comments

Figure 1

1) Although the authors provide a reason for the selection of ZT11 and ZT23 as time-points to infect, sampling evenly around the clock should be done to map a circadian

response for at least one readout. This could be done by recording weight loss at 7-8 dpi when infections are conducted at ZT5, ZT11, ZT17, and ZT23.

Response: We have conducted new experiments and included ZT5 and ZT17 as separated points for infection. The results of the experiment comparing the outcomes of mice infected 6hrly intervals at ZT5, ZT11, ZT17 and ZT23 revealed that ZT11 had the highest and that ZT23 had the lowest mortality. Therefore, we validate the choice of ZT11 and ZT23 for all future experiments.

2) It would be better to show the confirmation experiment using reversed light cycles as a supplemental figure rather than state data not shown (line 83).

Response: We have done so in supplementary Figure 2.

3) Regarding panels 1E and F, mice are reported to be kept in constant darkness for 1 week prior to infection at different CTs. These are valuable experiments but keep in mind that because of free running, mice in this experiment likely experienced a 3.5 hour phase advance in the week leading up to influenza infection. Thus, the time points being compared are more akin to ZT7.5 and ZT19.5 (assuming an intrinsic period of 23.5 hours).

Response: We agree with the reviewer. The experiments were done with animals kept in darkness for 3-7 days prior to infections. Only in one case, did we extend to 7days, we have removed those data and included the revised data in our figures. We have corrected the methods to reflect these changes.

Figure 2

1) It would be better to show the gender segregated data for influenza infection as a supplemental figure panel rather than reporting it as data not shown (line 138).

Response: We have done so for figure S3 in the supplement. We do not have data for female animals at 6hr and 12hrs post-infection.

Figure 3

1) This figure would be enhanced by some targeted cytokine measurements to validate the interpretation of gene expression data that ZT11 infections lead to lung hyper-inflammation.

Response: We have assayed early time points following infection in both whole lung and on day 6 in bronchoalveolar lavage. We have reported the same in Fig 3 and Fig S9. However, barring the exception of IL10, no differences achieved statistical significance. There could be a few possibilities for these observations. First, it is well known that the cytokine differences follow the transcriptomic and viral titer peaks. Therefore, the differences are probably more evident on day 8-10 following infection. However, since our goal, was to uncover the early differences that mediate circadian gating of IAV infection, we only looked at days 1-6 instead. Second, it is also possible that the difference in

outcome is not a result of absolute differences in cytokine expression rather secondary to how the lung parenchyma responds to the same levels of cytokines in the ZT 23 versus ZT11 group. We have shown that circadian control is mediated by the club cells in the lung, lending support to this latter hypothesis. In any case, it is intriguing to have such differences in the outcome but not an early difference in the pro-inflammatory cytokines.

Figure 4

1) For panel 4B please provide alveolar MAC counts as a proportion of live cells. Influenza infection ought to lead to a depletion of this cell type based on prior literature, not an increase.

Response: Thank you for the comment. We have corrected this figure. The reviewer is right in pointing out that the alveolar Macs should decrease with infection.

2) For panels 4D and 4E please include NKT and NK cell counts for uninfected mice at ZT11 and ZT23. Without this baseline data it is hard to interpret the difference in cell counts observed at on the first day post-infection.

Response: We have included these data in the supplementary figures (Fig. S11A).

3) Please provide control data demonstrating the extent of NK cell depletion by antibody treatment at ZT23 vs ZT11 (based on panel 4D, 1 dpi appears to be the time with the greatest differential cell count).

Response: We have included these data in the supplementary figures.

Figure 5

1) The paper would be enhanced by checking NK/NKT/inflammatory monocyte cell number in the conditional *bmal1* knockout lines and/or the *bmal1* iKO mice. This would mechanistically tie together the observations in Figure 4 and Figure 5 and make for stronger conclusions. As the paper currently stands, the idea that NK cell number mediates circadian effects during influenza is not validated and is based on a single experimental approach.

Response: We thank the reviewer for this comment. We have accordingly added the results from iKO animals not only for the flow cytometry data, but also for the viral titers and the lung histology. We found that there was no time of day difference in the viral titers and histology in the Cre^+ animals. For the flow cytometry, we found that the % of NK1.1+ cells were no different in the Cre^+ group, while they were higher in the CT23 group in Cre^- littermates. Even though the inflammatory monocytes seemed higher in the $Ercre^+$ animals, the differences failed to attain reach statistical significance.

Miscellaneous

1) Line 152: Can you confirm reference 22 is correct for citing circadian variation in lung leukocyte trafficking?

Response: Thank you for this comment. Checked and corrected accordingly.

Reviewer #2 (Remarks to the Author):

Summary

There is emerging evidence that circadian rhythms pervade many physiological responses in many tissues, including, importantly, inflammatory responses to pathogens. It is of interest to understand the immune mechanisms by which circadian rhythms affect outcomes. In a murine model of influenza A virus infection, the authors conduct a detailed investigation of the effects of timing of initial inoculum with respect to the circadian rhythm. They provide evidence that initial infection just before the start of the active period leads to higher mortality. They then dissect the underlying mechanisms to show this is not due to a difference in viral titre but an overexuberant inflammatory response, specifically including increased Ly6chi inflammatory monocytes and decreased NK1.1+ NK cells and iNKT cells. Finally, using specific deletion of clock genes in myeloid cells or pulmonary club cells they show outcomes are influenced both by circadian rhythms in myeloid cells and in epithelial cells.

This paper addresses an important, topical question and is well written, with a logical series of elegant experiments.

Major criticisms

1. The data presented appear solid, with reasonable numbers of mice and replications of experiments. However the manuscript does not mention blinding. In particular this can affect judgements about when to cull marginal animals with severe disease, and even more so clinical scoring undertaken in Fig 1C. Please state whether any blinding was used in the study (lack of blinding should not preclude publication, but should be acknowledged as a limitation). Please state whether any blinding was used for the histological scoring, as would be standard).

Response: The outcomes were all assessed in a blinded fashion. The researchers who weighed and scored the animals, were unaware of the treatment group assignment or genotype. We have modified the methods to reflect the same.

2. Related to this, the key foundational observation is expressed in Figure 1. Accurate weighing is essential for this readout. The mice were weighed once daily and so there may be some intrinsic diurnal variation in weight which could tend to bias risk of mortality in favour of one group more than another (one group will be weighed at the nadir of their daily weight variation). In the ideal experiment the mice would have been weighed twice daily, in a blinded fashion. Please discuss, and/or provide evidence of the magnitude of diurnal variation in weight during illness.

Response: We thank the reviewer for this comment that has given us an opportunity to clarify our experimental design further. We weigh animals once a day, at 24hr intervals from the time of the last weight. The first weight is at the time of infection and serves as the baseline. Therefore, as the reviewer has rightly pointed out, one group is being weighed at ZT 23 and another at ZT11. However, once the animals reach 20% body weight loss, we start weighing them twice daily, such that each group is being weighed at ZT23 and ZT11 (or CT23 and CT11), so ensuring that we account for potential differences with diurnal variation. In the additional experiments, conducted for this revision (that included infections at ZT5, ZT17, ZT23 and ZT11), we weighed animals twice daily as recommended; this did not change overall outcomes as shown in Figure S2. We have clarified the description of our experimental design in the methods section.

3. The transcriptomic experiment is potentially interesting, but the analysis performed is rather limited leading to bland/uninformative conclusions. At least tables of differentially expressed genes should be available in online supplement. Would suggest using GSEA for signatures of specific pathways of interest, eg TLR signalling. Indeed, GSEA is mentioned in the methods, but no analysis is shown.

Response: We have added more pathway analyses data in the supplement and updated the methods to reflect the same. The tables of the differentially expressed genes are also included now in the supplementary section.

4. Discussion: this work has significant potential clinical relevance eg. for shift workers, management of flu pandemics, likely extension to other respiratory pathogens etc. Currently this is restricted to two sentences: please expand discussion of the likely implications of these findings.

Response: We thank the reviewer for this opportunity to expand on the implications of our study. We have done so.

Minor criticisms

1. Results: line 65 specify dose of PR8 here, as well as in methods. **Done**
2. Results: line 83 and line 138 state 'Data not shown'. Please show data in an online supplement or remove. **Added supplementary figures to support the same.**
3. Results line 96, a change in the magnitude of an effect cannot be attributed to a sample size. Indeed, might it not be more probable that this difference between models was due to an effect of light signal. Please change this explanation / if you believe this is a sample size problem, suggest perform further replicates.

Response: Thank you for this comment. We have accordingly revised our explanation for the same.

4. Results paragraph beginning line 169. Please specify more about the design of this experiment: eg number of replicates (it is mentioned in methods but should also be mentioned here) **Done.**

5. Results line 195: qualify the magnitude: '..at ZT11 had slightly higher total CD45+...' **Done**

6. Results line 200 and following: I wonder about correction for multiple comparisons. How many cell types were compared? Was there an *a priori* hypothesis? What is the risk here of Type I statistical error?

Response: We thank the reviewer for this comment. Our a priori hypothesis included innate immune cells of the type that are listed in the results. This is how we generated the flowcytometry panel. We did not include adaptive immune cells in our panels and other than NK1.1+ cells, there weren't any others from the lymphocytic lineage. We have however, included post-hoc Bonferroni correction for multiple comparisons for both flow and cytokine data.

7. Discussion line 261 ff: Could you speculate on what might be the evolutionary advantage of the clock signal enhancing inflammation at certain periods.

Response: We thank the reviewer for this comment and the opportunity to improve our discussion. We have accordingly added the same to our discussion.

8. Line 276: Please add the citation numbers after 'Edgar' 'Ehlers' line 284, please add relevant citation for this statement. **Done**

9. Minor typos: line 196 remove extraneous comma, 409 'were' not 'was' **Done**

10. Figure 2A: Legend: was this a single iteration of the experiment or were there more than one replicate performed?

Response: Data shown were pooled from 3-4 independent experiments. We have updated the legend accordingly.

11. Figure 2: red squares are hard to differentiate when overlapped. Suggest use diamonds instead, and make the median lines more visible (eg red group could have black summary

statistics) **We have updated the figure accordingly.**

12. **Figure 2C and 2D, 4D:** bars hide data. Please show individual data points, eg as an overlay over the bars, rather than solely summary statistics. **Done**

13. **Figure 3A:** Specify if there was any blinding: **Yes, the researcher scoring the sections was blinded to the study assignment. We have likewise updated in methods and legends.**

14. **Figure 4A:** y axis should stop at (or a little above) 100%, not to 150%

15. **Figure 4B:** the gating looks messy: why are there black gates and red gates. Which was actually used for the analysis? Please show only the relevant data, or if separate gates were used for ZT23 and ZT11 groups, please explain.

Response: We have corrected the same. The red gate was placed to highlight the population better. All analyses was done based on the black gates which were identical for the ZT11 and ZT23 groups.

16. **Fig 4B and 4C:** axes are untidy and could be better formatted. **Done.**

17. **Methods:** statistics section needs a statement on reproducibility (ie number of replicates of experiments.) **Done.**

18. **References:** are incomplete for refs 10,13,38,44

Response: We have corrected the same.

Overall appraisal

An interesting manuscript presenting a systematic and logical series of experiments.

Reviewer #3 (Remarks to the Author):

A manuscript by Sengupta et al described the effects of circadian rhythms on influenza virus infection. The authors found differences in influenza-induced morbidity and survival, delayed viral clearance in ZT11 mice, associated with increased lung inflammation and pathology. Differences in NK1.1 and monocyte percentages are reported.

These are interesting findings but the data need to be re-analysed before any conclusions can be made. My main concern is that some of the conclusions are based on either small differences in numbers or frequencies alone:

Fig 4A numbers: What are the actual differences in cell numbers for d1, d2 and d3 significant differences? It seems to me as perhaps 80 cells? Would these be physiologically significant differences? How many mice were analysed? How many times was the experiments repeated? Showing pulled data across experiments is needed to draw conclusions here.

Response: We thank the reviewer for this comment. The difference between the two groups on days 2 and 4 specifically is in the order of $150-200 \times 10^4$ cells, which should result in biologically meaningful differences, as supported by the difference in mortality. However, the Figure 4 A, particularly the $CD45^+$ cell numbers were harder to read due to the massive influx of cells in both groups on day 8. We have therefore modified the graph to remove the day 8 and 10 data (which were not relevant to our present experimental design or question in any case). We hope that this helps clarify the message. The experiment was repeated 3 times, with 3-6 animals per group each time. Further these results are also consistent with our BAL data, wherein the cell counts were higher in the ZT11 group on day 2 and 6 post-infection.

Fig 4B: numbers need to be shown for inflammatory monocytes to support the conclusions. Also, statistical data should be checked as d6 does not look statistically significant.

Response: We are grateful to the reviewer for drawing our attention to this problem. We have added the number of inflammatory monocytes alongside the one representing them as a percentage of $CD45^+$ cells. The statistical testing has also been checked and appropriately corrected.

Fig 4C: numbers should also be shown as this is what matters with respect to inflammation at the site of infection.

Response: We have added the same and revised the text of the manuscript in the related portion.

Fig 4D: also numbers need to be shown. NKT cells: significant differences are shown for 0.2%. Is this physiologically significant? Are the numbers different?

Response: We thank the reviewer for this question. The numbers for the $NK1.1^+$ are not different. We have added these data in Fig 4D. Originally this is why we used the $Nk1.1$ antibody to deplete this population, testing whether these cells are relevant to the circadian control of IAV infection. During our revision, we have repeated the $NK1.1$ depletion (followed by infection) twice more to confirm our findings. The revised data are reflected in Fig 5(A) and confirm that upon depleting $NK1.1^+$ cells, the time of day difference is abolished. Since we didn't employ a specific strategy to deplete NKT cells, we agree with the reviewer that it is hard to justify the physiological relevance of this cell type.

Other comments:

Figure 1: the authors mention that infecting mice in reverse light-dark cycles confirm results in Fig 1. These are very important controls as should be shown in Fig. 1.

Response: We thank the reviewer for this comment and have accordingly added the same to the supplementary figures.

Reviewers' Comments:

Reviewer #1:

Remarks to the Author:

The authors have thoroughly addressed my concerns. Regarding the newly added data, I had trouble interpreting Figure S8 but feel this can be addressed at the editing level.

Reviewer #2:

Remarks to the Author:

The authors have addressed all of the concerns I raised.

I note that the numbering of supplemental figures is not correct in the supplemental figure legends - please check - and there is a typo ('if' not 'of') in figure S7/8 legend.

Reviewer #3:

Remarks to the Author:

The manuscript has been improved considerably by addressing the reviewers' comments.